# Solution of A Complex Nonlinear Fractional Biochemical Reaction Model

Fatima Rabah, Marwan Abukhaled * and Suheil A. Khuri

Department of Mathematics and Statistics, American University of Sharjah, Sharjah P.O. Box 26666, United Arab Emirates; g00049347@alumni.aus.edu (F.R.); skhoury@aus.edu (S.A.K.)

* Correspondence: mabukhaled@aus.edu

**Abstract:** This paper discusses a complex nonlinear fractional model of enzyme inhibitor reaction where reaction memory is taken into account. Analytical expressions of the concentrations of enzyme, substrate, inhibitor, product, and other complex intermediate species are derived using Laplace decomposition and differential transformation methods. Since different rate constants, large initial concentrations, and large time domains are unavoidable in biochemical reactions, different dynamics will result; hence, the convergence of the approximate concentrations may be lost. In this case, the proposed analytical methods will be coupled with Padé approximation. The validity and accuracy of the derived analytical solutions will be established by direct comparison with numerical simulations.

**Keywords:** enzyme inhibitor; biochemical reaction; fractional differential system; Laplace transformation; semi-analytic

## 1. Introduction

Data gathering and experimental analysis do not generally provide rigorous tools for understanding the kinetics of modern complex physical, biological, and biochemical research. Therefore, researchers have increasingly employed mathematical modeling, where theoretical analysis would lead to new insights and pave the way for better designs and controlled systems [1–6].

A desired feature of fractional operators is their essential multiscale nature. Consequently, time-fractional operators empower memory effects. In other words, the response of a system is dependent on its previous history. In contrast, space-fractional operators enable nonlocal and scale effects [7]. This nonlocal property of fractional derivatives gives insight into a system's future state features from the previous and present states. Therefore, fractional models are more suitable for simulating physical phenomena and hence more accurate for biochemical reactions. Moreover, fractal geometries that model nonlocal transport, which arises in complex microstructural systems, are often seen in fractional derivative models [8].

Recent research has affirmed that modeling natural phenomena arising in biology, chemistry, and physics with fractional differential equations is more suitable for describing memory and hereditary properties of various materials and processes. For example, Ionescu et al. detailed, in a comprehensive review, the latest developments in fractional calculus applications in biological systems [9]. Rihan discussed some fractional-order differential models of biological systems with memory, such as dynamics of tumor-immune system and dynamics of HIV infection [10]. Other examples of fractional models covering various fields of sciences and engineering can be found in fluid flow [11], electrical networks [12], viscoelasticity [13], and control theory [14]. The reader is encouraged to see the recently published survey-cum-expository review article [15], and the following articles, which shed more light on the discussion on and applications of fractional models [16–21].

Nonetheless, exact solutions to most nonlinear fractional-order differential equations cannot be found. Therefore, many semi-analytical and numerical methods have been

developed in recent years to find approximate solutions instead. Most classical numerical methods used for ordinary differential equations have been successfully modified for fractional differential equations such as implicit Euler scheme [10], spectral collocation methods [22], Adams–Bashforth methods [23], and Runge–Kutta methods [24]. Some of the newly developed numerical methods include a new predictor-corrector formula, Legendre spectral method, discretization of Riemann–Liouville, and a modified Adams–Bashforth method [25–28].

Although numerical solutions can be accurate and efficiently obtained, they have some drawbacks that make them less appealing than analytical solutions. Numerical stability and adjusting parameters to match the numerical data can be exceptionally challenging [29]. As with numerical methods, most analytical schemes that have been initially developed for integer-order differential systems have been modified for fractional differential systems [30–36].

This paper studies a nonlinear fractional model of enzyme inhibitor reactions subject to two different sets of initial conditions and kinetic parameters. Modified Laplace decomposition and differential transformation methods are applied to derive simple analytical expressions for the concentrations of species. The obtained expressions converge and stabilize over a prescribed small time domain. However, with possible divergent solutions over large intervals, these methods are coupled with Padé approximation to maintain convergent series solutions for larger reaction times [37]. The used methods are accessible to the broader research community and can be adapted to solve other models that arise in chemistry and chemical engineering.

## 2. A Model of Complex Enzyme Inhibitor Reactions

Consider the complex chemical reaction network for mixed enzymatic inhibition as shown in Figure 1.

$$\mathcal{E} + S \underset{k_2}{\overset{k_1}{\rightleftharpoons}} \mathcal{E}S \overset{k_3}{\longrightarrow} \mathcal{E} + \mathcal{P}$$

**Figure 1.** A complex chemical reaction for a mixed enzymatic inhibition.

Where $\mathcal{E}, S, \mathcal{P}$, and $\mathcal{I}$ represent enzyme, substrate, product, and inhibitor, respectively. $\mathcal{E}S, \mathcal{E}I$, and $\mathcal{E}SI$ represent the complex intermediate species. The parameters $k_1, \cdots, k_9$ represent the rate constants. If we express the concentrations of $\mathcal{E}, S, \mathcal{P}, \mathcal{I}, \mathcal{E}S, \mathcal{E}I$, and $\mathcal{E}SI$ by $E, S, P, I, C_1, C_2$, and $C_3$, respectively, then the mass action law leads to the following nonlinear fractional differential model, which is a modification of the integer-derivative model discussed by Akgül et al. [38]:

$$
\begin{aligned}
D_t^\alpha E &= -k_1 ES + (k_2 + k_3)C_1 - k_4 EI + k_5 C_2 S, \\
D_t^\alpha S &= -k_1 ES + k_2 C_1 + k_4 EI - (k_5 + k_8)C_2 S + k_9 C_3, \\
D_t^\alpha I &= -k_4 EI + k_5 C_2 S - k_6 C_1 I + k_7 C_3, \\
D_t^\alpha P &= k_3 C_1, \\
D_t^\alpha C_1 &= k_1 ES - (k_2 + k_3)C_1 - k_6 C_1 I + k_7 C_3, \\
D_t^\alpha C_2 &= k_4 EI - (k_5 + k_8)C_2 S + k_9 C_3, \\
D_t^\alpha C_3 &= k_6 C_1 I - (k_7 + k_9)C_3 + k_8 C_2 S,
\end{aligned}
\tag{1}
$$

where $0 < \alpha \le 1$. $D_t^\alpha$ is the Caputo fractional derivative defined by

$$
D_{x_0}^\alpha f(x) = {}^{\text{RL}}D_{x_0}^\alpha \left( f(x) - \sum_{k=0}^{m-1} \frac{f^{(k)}(x_0)}{k!}(x - x_0)^k \right),
\tag{2}
$$

where ${}^{\text{RL}}D_{x_0}^\alpha f(x) = D^m\left(J_{x_0}^{m-\alpha} f(x)\right)$, $m - 1 < \alpha \le m$, and $m \in \mathbb{N}$, and $J_{x_0}^\alpha f(x)$ is the Riemann–Liouville fractional integration of order $\alpha$ for a real-valued function $f : \mathbb{R}^+ \to \mathbb{R}$ defined by

$$
J_{x_0}^\alpha f(x) = \frac{1}{\Gamma(\alpha)} \int_{x_0}^x (x - s)^{\alpha - 1} f(t)\,dt,\ \alpha > 0, x > 0.
\tag{3}
$$

## 3. Analytical Expressions for the Concentrations

Consider the nonlinear fractional reaction system (1) subject to the following set of initial concentrations:

$$
E(0) = e_0,\ S(0) = s_0,\ I(0) = i_0,\ P(0) = p_0,\ C_1(0) = c_{1_0},\ C_2(0) = c_{2_0},\ C_3(0) = c_{3_0}.
\tag{4}
$$

We will derive two approximate analytical expressions of the concentrations of enzyme, substrate, product, inhibition, and the complex intermediate species using modified Laplace decomposition (LDM) and differential transformation (DTM) methods.

The difference between Riemann–Louivelle and Caputo fractional derivatives, which is just in the order of operators, makes Caputo definition closer to the traditional integer-derivative operator and hence more used than Riemann–Louivelle.

### 3.1. Laplace Decomposition Approach

We begin with the following lemma whose proof follows immediately from (2) and (3) [39].

**Lemma 1.** *The Laplace transform of the Caputo fractional derivative of order $\alpha$ is given by*

$$
\mathcal{L}\{D^\alpha f(x)\} = \frac{s^m F(s) - \sum_{i=1}^m s^{m-i} f^{(i-1)}(0)}{s^{m-\alpha}},
\tag{5}
$$

*where $m \in \mathbb{N}$ and $m - 1 < \alpha \le m$.*

Applying Laplace transform to each equation in the reaction system (1) gives

$$
\begin{aligned}
\mathcal{L}\{E(t)\} &= \frac{e_0}{s} + \frac{1}{s^\alpha}\mathcal{L}\{-k_1 ES + (k_2 + k_3)C_1 - k_4 EI + k_5 C_2 S\}, \\
\mathcal{L}\{S(t)\} &= \frac{s_0}{s} + \frac{1}{s^\alpha}\mathcal{L}\{-k_1 ES + k_2 C_1 + k_4 EI - (k_5 + k_8)C_2 S + k_9 C_3\}, \\
\mathcal{L}\{I(t)\} &= \frac{i_0}{s} + \frac{1}{s^\alpha}\mathcal{L}\{-k_4 EI + k_5 C_2 S - k_6 C_1 I + k_7 C_3\}, \\
\mathcal{L}\{P(t)\} &= \frac{p_0}{s} + \frac{1}{s^\alpha}\mathcal{L}\{k_3 C_1\}, \\
\mathcal{L}\{C_1(t)\} &= \frac{c_{10}}{s} + \frac{1}{s^\alpha}\mathcal{L}\{k_1 ES - (k_2 + k_3)C_1 - k_6 C_1 I + k_7 C_3\}, \\
\mathcal{L}\{C_2(t)\} &= \frac{c_{20}}{s} + \frac{1}{s^\alpha}\mathcal{L}\{k_4 EI - (k_5 + k_8)C_2 S + k_9 C_3\}, \\
\mathcal{L}\{C_3(t)\} &= \frac{c_{30}}{s} + \frac{1}{s^\alpha}\mathcal{L}\{k_6 C_1 I - (k_7 + k_9)C_3 + k_8 C_2 S\}.
\end{aligned}
\tag{6}
$$

We seek an approximate solution to system (6) and hence a solution to the fractional system (1) in the form of a power series about $t = 0$, that is

$$E(t) = \sum_{n=0}^{\infty} E_n(t), \quad S(t) = \sum_{n=0}^{\infty} S_n(t), \quad I(t) = \sum_{n=0}^{\infty} I_n(t), \quad P(t) = \sum_{n=0}^{\infty} P_n(t),$$

$$C_1(t) = \sum_{n=0}^{\infty} C_{1_n}(t), \quad C_2(t) = \sum_{n=0}^{\infty} C_{2_n}(t), \quad C_3(t) = \sum_{n=0}^{\infty} C_{3_n}(t). \tag{7}$$

The nonlinear terms in system (6) are expressed in terms of Adomian polynomials as follows:

$$ES = \sum_{n=0}^{\infty} A_{1n} = \frac{1}{n!} \left( \frac{d}{d\lambda} \right)^n \left( \sum_{k=0}^{n} \lambda^k E_k \sum_{k=0}^{n} \lambda^k S_k \right) \Big|_{\lambda=0},$$

$$EI = \sum_{n=0}^{\infty} A_{2n} = \frac{1}{n!} \left( \frac{d}{d\lambda} \right)^n \left( \sum_{k=0}^{n} \lambda^k E_k \sum_{k=0}^{n} \lambda^k I_k \right) \Big|_{\lambda=0},$$

$$C_1 I = \sum_{n=0}^{\infty} A_{3n} = \frac{1}{n!} \left( \frac{d}{d\lambda} \right)^n \left( \sum_{k=0}^{n} \lambda^k (C_1)_k \sum_{k=0}^{n} \lambda^k I_k \right) \Big|_{\lambda=0},$$

$$C_2 S = \sum_{n=0}^{\infty} A_{4n} = \frac{1}{n!} \left( \frac{d}{d\lambda} \right)^n \left( \sum_{k=0}^{n} \lambda^k (C_2)_k \sum_{k=0}^{n} \lambda^k S_k \right) \Big|_{\lambda=0}. \tag{8}$$

Substituting (7) and (8) recursively in (6) and then applying the inverse Laplace transforms lead to the analytical expressions of all concentrations expressed in series forms. The first two terms of each of these series are given below

$$E_0 = e_0, \; S_0 = s_0, \; I_0 = i_0, \; P_0 = p_0, \; C_{1_0} = c_{1_0}, \; C_{2_0} = c_{2_0}, \; C_{3_0} = c_{3_0},$$

$$E_1 = (s_0 c_{2_0} k_5 - s_0 e_0 k_1 + c_{1_0} k_2 + c_{1_0} k_3 - i_0 e_0 k_4) \frac{t^\alpha}{\Gamma(\alpha + 1)},$$

$$S_1 = (-s_0 c_{2_0} k_5 - s_0 c_{2_0} k_8 - s_0 e_0 k_1 + c_{1_0} k_2 - c_{3_0} k_9 + i_0 e_0 k_4) \frac{t^\alpha}{\Gamma(\alpha + 1)},$$

$$I_1 = (s_0 c_{2_0} k_5 - i_0 c_{1_0} k_6 + c_{3_0} k_7 - i_0 e_0 k_4) \frac{t^\alpha}{\Gamma(\alpha + 1)},$$

$$P_1 = (c_{1_0} k_3) \frac{t^\alpha}{\Gamma(\alpha + 1)}, \tag{9}$$

$$C_{11} = (-s_0 e_0 k_1 - c_{1_0} k_2 - c_{1_0} k_3 - i_0 c_{1_0} k_6 + c_{3_0} k_7) \frac{t^\alpha}{\Gamma(\alpha + 1)},$$

$$C_{21} = (-s_0 c_{2_0} k_5 - s_0 c_{2_0} k_8 + c_{3_0} k_9 + i_0 e_0 k_4) \frac{t^\alpha}{\Gamma(\alpha + 1)},$$

$$C_{31} = (s_0 c_{2_0} k_8 + i_0 c_{1_0} k_6 - c_{3_0} k_7 - c_{3_0} k_9) \frac{t^\alpha}{\Gamma(\alpha + 1)}.$$

### 3.2. Differential Transformation Method

First proposed by Zhou [40], the differential transformation method (DTM) is an iterative approach for obtaining a Taylor series solution of a differential equation without the need for the tedious computing of symbolic higher derivatives. Arikoglu and Ozkol [41] modified the original version of the DTM to make it applicable to solve fractional differential equations. In this section, we derive a series solution of system (1) using the fractional DTM [42].

The fractional power series expansion of the continuous analytical function $f(x)$ is given by

$$f(t) = \sum_{k=0}^{\infty} F(k)(t - t_0)^{k/\alpha}, \tag{10}$$

where $F(k)$ is the fractional differential transformation of $f(t)$ defined by

$$
F(k) = \begin{cases} \frac{1}{(k/\alpha)!} D^{k/\alpha}\Big|_{t=t_0}, & \text{if } k/\alpha \in Z^+, k = 0, 1, 2, \ldots, (q\alpha - 1) \\ \\ 0, & \text{if } k/\alpha \notin Z^+ \end{cases} \tag{11}
$$

For a fractional-order $q$, the Caputo fractional derivative is given by

$$
D_{t_0}^q f(t) = \frac{1}{\Gamma(m-q)} D^m \left\{ \int_{t_0}^t \left[ \frac{f(t) - \sum_{k=0}^{m-1} (1/k!)(t-t_0)^k f^{(k)}(t_0)}{(t-x)^{1+q-m}} \right] dt \right\}. \tag{12}
$$

The following properties of fractional differential transformations are needed in the derivation of the analytical solution of system (1) [42].

**Theorem 1.** *If* $f(x) = g_1(x) \pm g_2(x) \pm \cdots \pm g_n(x)$, *then* $F(k) = G_1(k) \pm G_2(k) \pm \cdots \pm G_n(k)$.

**Theorem 2.** *If* $f(x) = \prod_{j=1}^n g_j(x)$, *then*

$$
F(k) = \sum_{k_{n-1}=0}^k \sum_{k_{n-2}=0}^{k_{n-1}} \cdots \sum_{k_2=0}^{k_3} \sum_{k_1=0}^{k_2} G_1(k_1) G_2(k_2 - k_1) \ldots G_{n-1}(k_{n-1} - k_{n-2}) G_n(k - k_{n-1}).
$$

**Theorem 3.** *If* $f(x) = (x - x_0)^p$, *then* $F(k) = \delta(k - \alpha p)$, *where*
$\delta(k) = \begin{cases} 1 & \text{if } k = 0 \\ 0 & \text{if } k \neq 0 \end{cases}$.

**Theorem 4.** *If* $f(x) = D_{x_0}^q[g(x)]$, *then* $F(k) = \frac{\Gamma(q+1+k/\alpha)}{T(1+k/\alpha)} G(k + \alpha q)$.

By applying the fractional operator in (12) to system (1), we obtain the same series solution given in (10) for the integer case. For fractional order derivatives, the variations between the LDM and DTM were very small, and will be discussed in the Results and Discussion section.

### 3.3. Padé Approximation

It is known that the convergence of the truncated series solutions obtained by Laplace decomposition and differential transformation methods are guaranteed only over small domains. The divergence of the series solution obtained by the LDM or DTM may also result for large initial conditions. In this case, LDM and DTM methods can be coupled with Padé approximation to insure convergence. The Padé approximant of the function $f(x)$, which is a convergent ratio of two polynomials constructed from its Taylor series expansion, gives a better approximation of the function, especially when there are poles.

When the function $f(x)$ is expressed as a power series, the $[L/M]$ Padé approximant is given by

$$
f(x) = \frac{P_L(x)}{Q_M(x)} = \frac{p_0 + p_1 x + p_2 x^2 + p_3 x^3 + \cdots + p_L x^L}{1 + q_1 x + q_2 x^2 + q_3 x^3 + \cdots + q_M x^M}. \tag{13}
$$

### 4. Results and Discussion

In this section, two study cases are presented. In each case, the nonlinear reaction system (1) is solved for a different set of parameters and a different set of initial concentrations.

**Example 1.** *To verify the accuracy of the proposed approaches, we first solve the underlined system for the integer-derivative,* $\alpha = 1$, *subject to the following initial conditions (4).*

$$e_0 = 0.1, \ s_0 = 0.2, \ i_0 = 0.01, \ p_0 = c_{10} = c_{20} = c_{30} = 0, \tag{14}$$

*and the following constant rates*

$$k_1 = 0.1, \ k_2 = 0.2, \ k_3 = 0.4, \ k_4 = 0.9, \ k_5 = 1, \ k_6 = 0.4, \ k_7 = 0.9, \ k_8 = 0.2, \ k_9 = 0.5. \tag{15}$$

*The LDM and DTM solutions were identical for all seven species. For example, the identical five-term series solution obtained by the LDM and the DTM representing the concentration of enzyme is given by*

$$E(t) = 0.1 - 0.0029\,t + 0.000778\,t^2 - 0.000152\,t^3 + 0.0000242\,t^4. \tag{16}$$

The analytical expressions of the concentrations of all other species are provided in the Supporting Information. Figure 2a–g shows that for the integer case ($\alpha = 1$), the derived analytical concentration curves obtained by the LDM and the DTM are identical and strongly agree with the fourth-order Runge–Kutta numerical curves. Figure 2 also reflects the temporal dependence of relative concentrations of enzyme reaction components. It is noticed that concentrations of enzyme, substrate, and product decrease as time increases, whereas the concentrations of inhibitor, enzyme–substrate, enzyme–inhibitor, and enzyme–substrate–inhibitor increase with time.

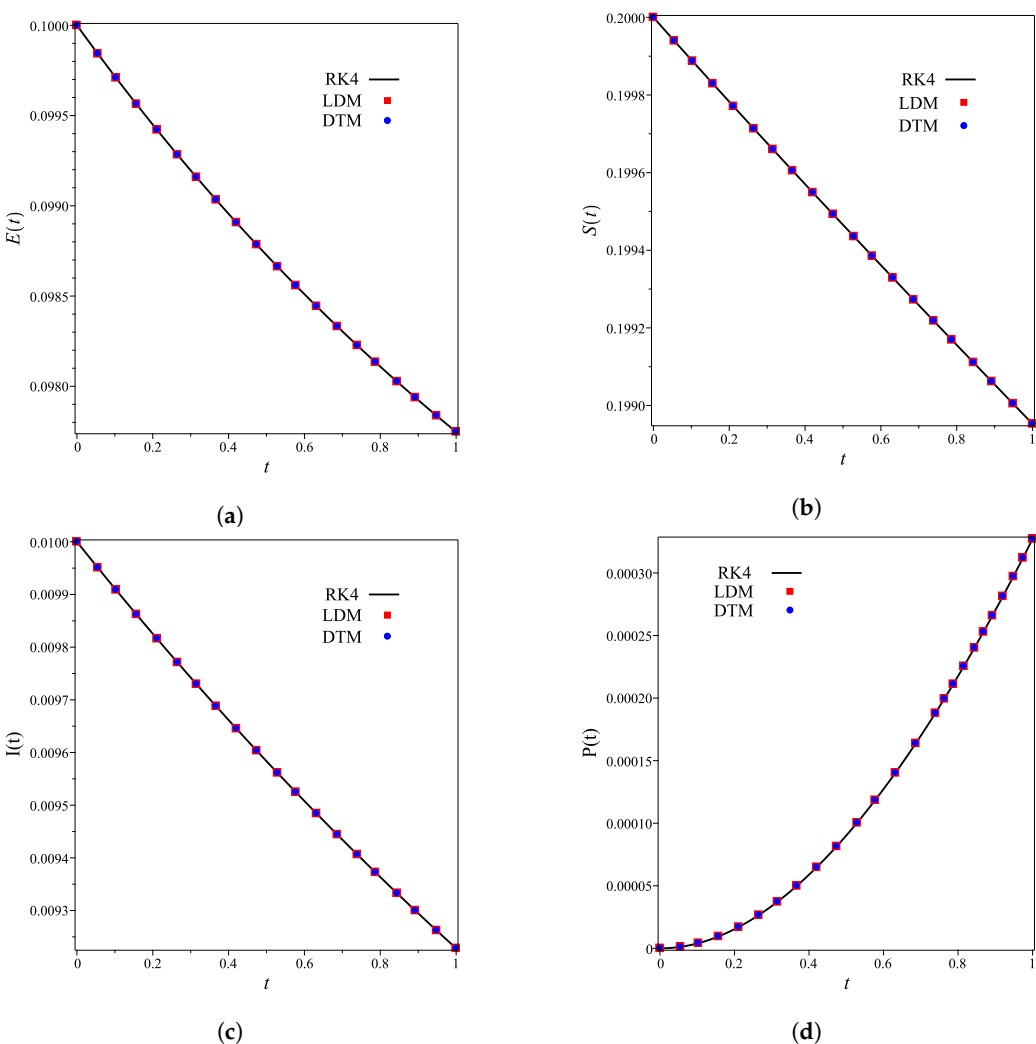

**Figure 2.** *Cont.*

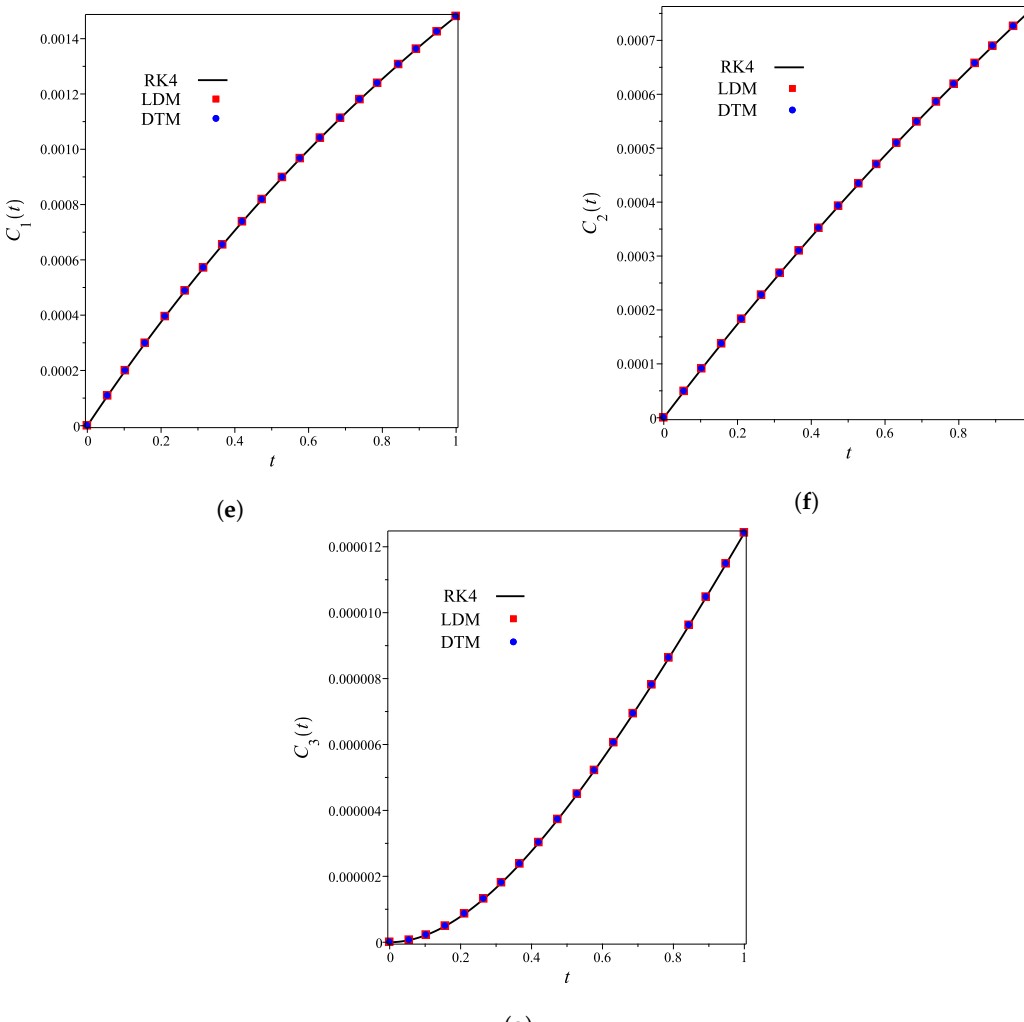

**Figure 2.** Analytical and numerical concentration curves for reaction system (1) for the integer-derivative case ($\alpha = 1$) with initial concentrations (14) and rate constants (15). (**a**) Enzyme. (**b**) Substrate. (**c**) Product. (**d**) Inhibitor. (**e**) Intermediate species ES. (**f**) Intermediate species EI. (**g**) Intermediate species ESI.

The nonlinear fractional reaction system (1) is also solved for the fractional derivatives $\alpha = 0.9$, and $\alpha = 0.8$. Figure 3a–g shows strong agreements between LDM and DTM concentration curves of enzyme, substrate, product, inhibition and all complex intermediate species. In this Figure, the fractional derivative $\alpha$ is an index of memory, where it is noticed that the concentrations of the enzyme components depend on the fractional order. Figure 3 clearly shows that as $\alpha$ increases, the fractional concentration curve gets closer to the curve representing the concentration for the integer case ($\alpha = 1$).

Figure 3a–c confirms that the enzyme, substrate, and inhibitor concentrations increase as the fractional power increases and decrease as time increases. In contrast, Figure 3d–g portrays that the product and intermediate species concentrations increase and reach their maximum with the rise of time and decrease of the fractional power.

Tables 1 and 2 assert that the actual variations between the LDM and DTM for the fractional cases are smaller than what they appear in Figure 3. This can also be inferred from the very small $y$-axis increments in Figure 3.

**Table 1.** Maximum variation between LDM and DTM computed concentrations when $\alpha = 0.9$.

| Concentration | Maximum Difference | Occurred at $x$ |
| --- | --- | --- |
| Enzyme | 0.0000424 | 1.000 |
| Substrate | 0.0000024 | 0.007 |
| Inhibition | 0.0000103 | 1.000 |
| Production | 0.0000205 | 0.925 |
| Complex ES | 0.0000323 | 0.925 |
| Complex EI | 0.0000105 | 1.000 |
| Complex ESI | 0.0000004 | 0.525 |

**Table 2.** Maximum variation between LDM and DTM computed concentrations when $\alpha = 0.8$.

| Concentration | Maximum Difference | Occurred at $x$ |
| --- | --- | --- |
| Enzyme | 0.0000843 | 0.850 |
| Substrate | 0.0000049 | 0.600 |
| Inhibition | 0.0000206 | 1.000 |
| Production | 0.0000408 | 0.825 |
| Complex ES | 0.0000643 | 0.825 |
| Complex EI | 0.0000208 | 1.950 |
| Complex ESI | 0.0000009 | 0.450 |

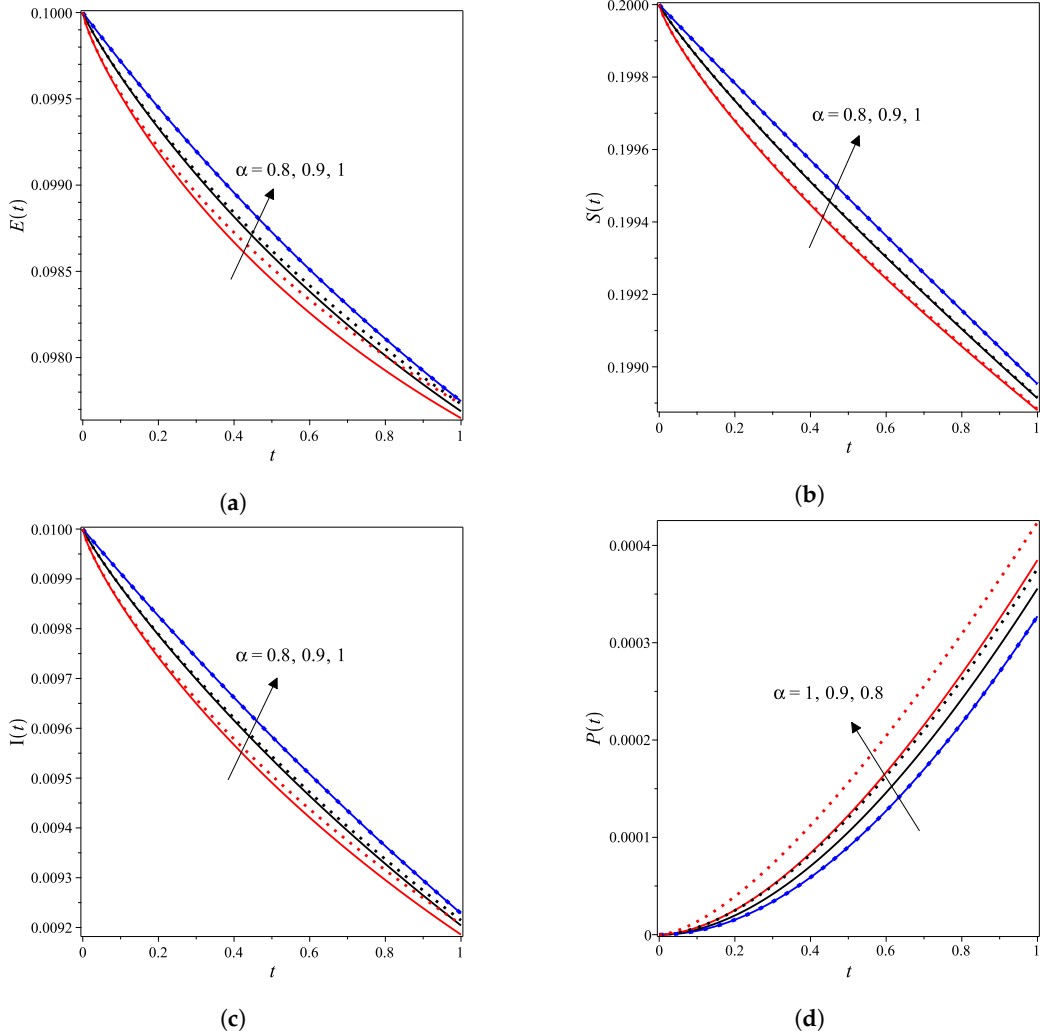

**Figure 3.** *Cont.*

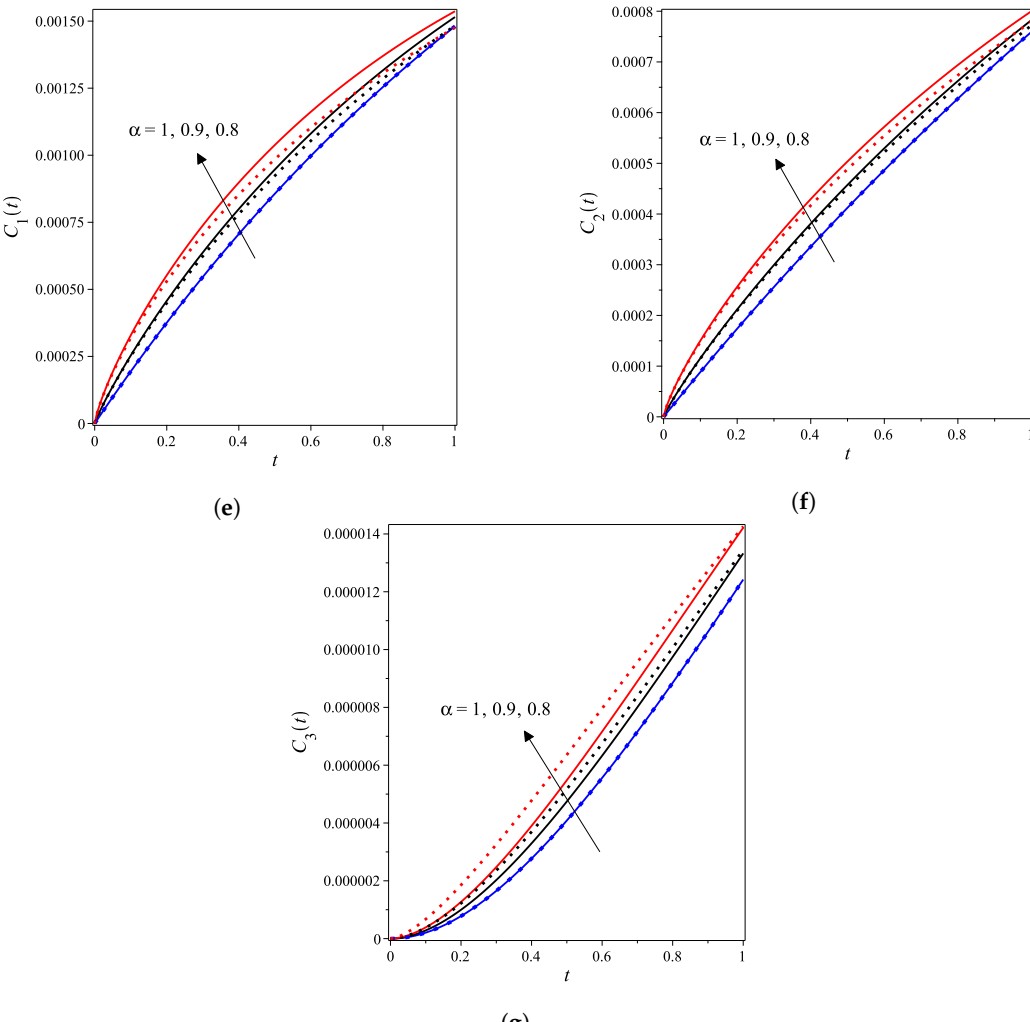

**Figure 3.** Analytical concentration curves for fractional reaction system (1) with initial concentrations (14) and rate constants (15) for the integer-derivative case $\alpha = 1$ and the fractional-derivative cases $\alpha = 0.9$ and 0.8. Solid and dotted curves represent the LDM and the DTM solutions, respectively. (**a**) Enzyme. (**b**) Substrate. (**c**) Product. (**d**) Inhibitor. (**e**) Intermediate species ES. (**f**) Intermediate species EI. (**g**) Intermediate species ESI.

**Example 2.** *Consider the nonlinear fractional reaction system (1) subject to the following set of relatively large initial concentrations:*

$$e_0 = 12, \; s_0 = 5, \; i_0 = 2, \; p_0 = c_{10} = c_{20} = c_{30} = 0, \tag{17}$$

*and the following set of constant rates*

$$k_1 = 0.01, \; k_2 = 0.2, \; k_3 = 0.04, \; k_4 = 0.19, \; k_5 = 0.1, \; k_6 = 0.4, \; k_7 = 0.09, \; k_8 = 0.22, \; k_9 = 0.05. \tag{18}$$

For the integer case, $\alpha = 1$, the obtained LDM, and DTM truncated series solutions (concentrations) were identical but diverged rapidly over a small domain. This divergence was controlled by using a [4/4] Padé approximant for each analytical derived expression. In Figure 4a, the divergent enzyme concentration curves obtained by the LDM and DTM are depicted against time. In contrast, Figure 4b shows how the use of Padé approximation overcomes this obstacle. Figure 5 is similar to Figure 4 but for the substrate concentration. All concentration curves for the case $\alpha = 1$ and their [4/4] corresponding Padé approximations are provided in the Supporting Information.

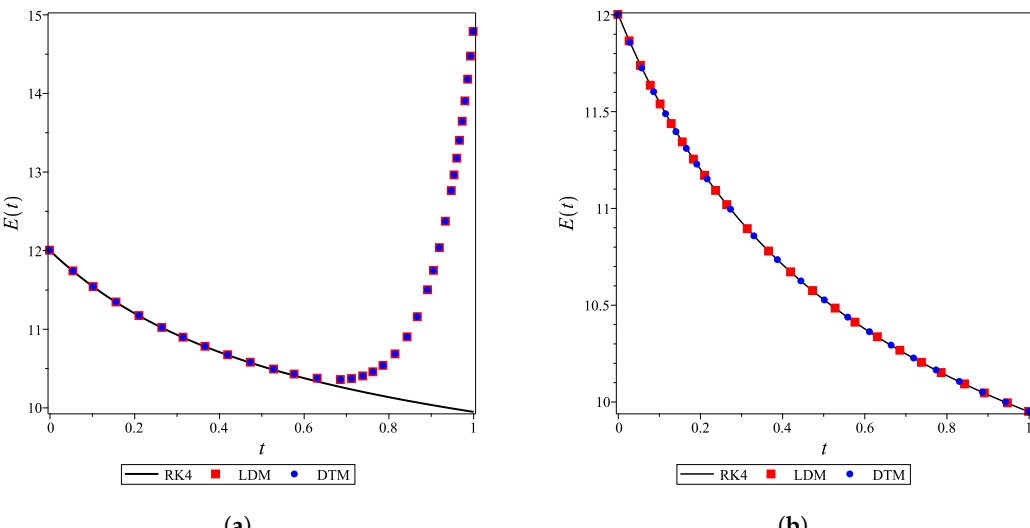

**(a)**  **(b)**

**Figure 4.** Analytical and numerical concentration curves of Enzyme ($E(t)$) for the integer-derivative system (1) with $\alpha = 1$, initial conditions (17), and parameters (18). (**a**) Divergent analytical concentration curve. (**b**) Convergent analytical concentration curve.

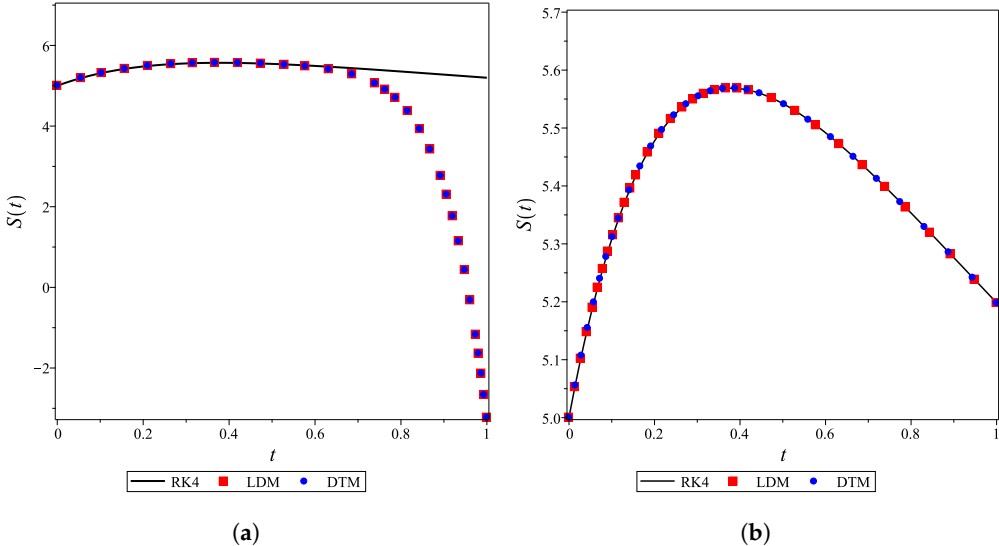

**(a)**  **(b)**

**Figure 5.** Analytical and numerical concentration curves of substrate ($S(t)$) for integer-derivative system (1) with $\alpha = 1$, initial conditions (17), and parameters (18). (**a**) Divergent analytical concentration curve. (**b**) Convergent analytical concentration curve.

The DTM was employed to derive analytical expressions for the concentration curves of all species for fractional values of $\alpha$ ($\alpha = 0.9, 0.8$). All the obtained curves of more than 10-term truncated series (provided in the Supporting Information) diverged over a relatively small domain. Therefore, large order Padé approximations were needed to obtain the convergent series solutions, as shown in Figure 6. A single command using Maple or MATLAB can be used to generate Padé approximations (given in supplementary material).

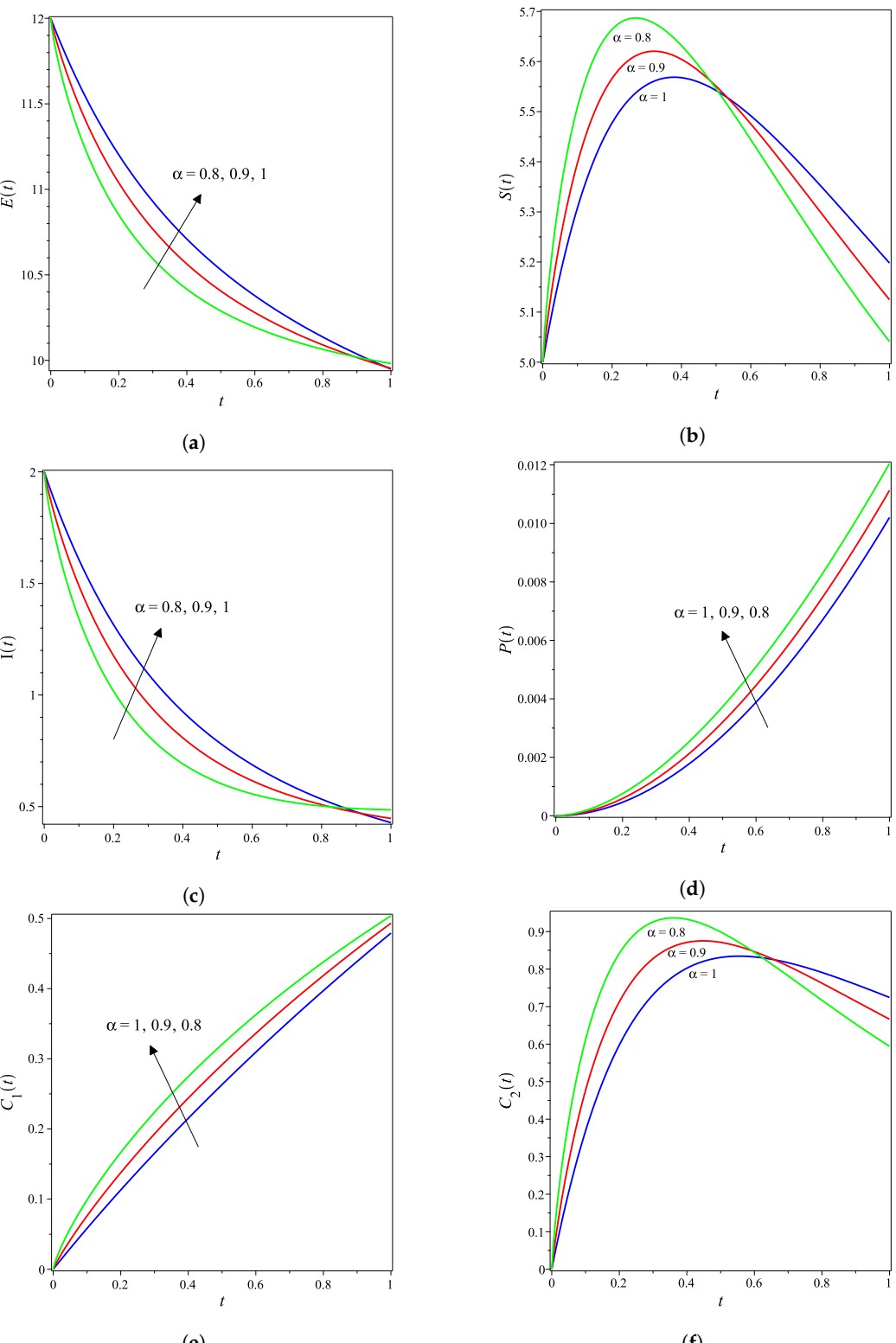

**Figure 6.** *Cont.*

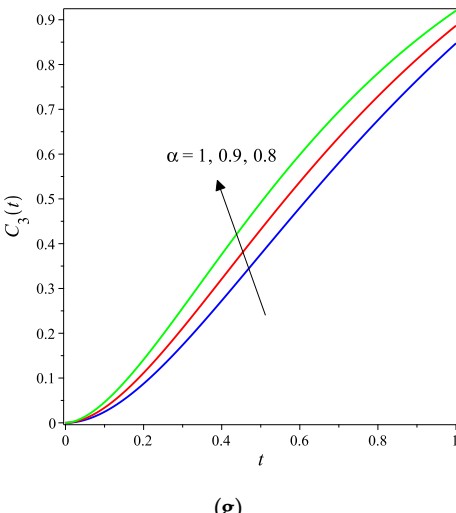

**(g)**

**Figure 6.** Analytical LDM concentration curves of $E, S, I, P, ES, EI$ and $ESI$ for system (1) with initial conditions (17) and parameters (18). (**a**) Enzyme. (**b**) Substrate. (**c**) Product. (**d**) Inhibitor. (**e**) Intermediate species ES. (**f**) Intermediate species EI. (**g**) Intermediate species ESI.

## 5. Conclusions

This paper discussed a complex nonlinear fractional model of enzyme inhibitor reactions subject to two different sets of initial concentrations, each with a different set of reaction rates. The simple, efficient, and reliable Laplace decomposition (LDM) and differential transformation (DTM) methods were utilized to solve the nonlinear fractional biochemical reaction system. The LDM was implemented by using Laplace transform of Caputo fractional derivative to convert the nonlinear fractional-derivative system (1) into an algebraic system, where the nonlinear terms are expressed in the form of Adomian polynomials. Then, the solution is obtained by employing the linearity of the Laplace and the inverse Laplace transforms. The fractional differential transformation method was implemented by directly applying Equations (10) and (11), and Theorem 2. The derived solution of system (1) represent the analytic expressions for the concentrations of enzyme, inhibitor, substrate, product, and the complex intermediate species: enzyme–substrate, enzyme–inhibitor, and enzyme–inhibitor–substrate were derived and discussed. From this study, it was concluded that different rate constants and initial concentrations produce different dynamics. Furthermore, it was shown that a Padé approximation of the series solution obtained by LDM and DTM would preserve convergence and stability when large initial concentrations or large rate constants are assumed. The derived LDM and DTM concentration expressions for the enzyme inhibitor reaction model were shown to be very close to the fourth-order Runge–Kutta method when the results were compared for the integer-derivative case.

The derived fractional analytical concentration curves would play a significant role in predicting the future state of the biochemical reaction model. In addition, the derived analytical expressions would be essential in investigating the effects of various reaction rates to reach better designs and controlled systems. The used methods are accessible to the broader research community. They can be extended to solve various fractional models to obtain a better insight into dynamical behavior for biological or chemical systems with possible hereditary properties.

**Supplementary Materials:** The following are available at https://www.mdpi.com/article/10.3390/mca27030045/s1.

**Author Contributions:** Conceptualization and methodology, M.A. and S.A.K.; software F.R.; validation, M.A., S.A.K. and F.R.; formal analysis, F.R. and S.A.K.; investigation, F.R.; writing—original draft preparation, F.R.; writing—review and editing, M.A.; visualization, S.A.K.; funding acquisition, M.A. All authors have read and agreed to the published version of the manuscript.

**Funding:** This work was partially supported by the American University of Sharjah Award #OAPCAS-1110-C00016. However, this paper represents the author's opinions and does not mean to represent the position or opinions of the American University of Sharjah.

**Conflicts of Interest:** The authors declare no conflict of interest.

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
