# Peer review of "Solution of a Complex Nonlinear Fractional Biochemical Reaction Model"

_mca, doi:10.3390/mca27030045_

Round 1

Author Response

Reply to Author 1 is attached.

Reviewer 2 Report

The authors discuss the problem of fractional order modelling of a biochemical process. The topic is for possible interest to the journal readers. The paper is well structured. The reference list is proper. The novelty of the paper must be detailed. It is used a fractional order for an existing model. Which is the physical meaning of the fractional derivative here? Which phenomena needs this calculus? The results obtained with Pade approximations must be compared with the original values in order to validate the approximation. In the Results section must be included the steps used to obtain these values. Please use a performance measure for each model in order to compare theme.

Author Response

Reply to Reviewer 2 has been uploaded

Reviewer 3 Report

In the paper "Theoretical analysis of a complex nonlinear fractional biochemical reaction model", nonlinear model of enzyme inhibitor rector was solved by two methods. The manuscript is well written. All aspects of model development are described in detail and are understandable to the readership. The manuscript can be published after minor revision according to the listed comments:

Numerical results would be useful in the abstract
The variables on the x and y axes should be marked in a larger font.

Author Response

Reply to Reviewer 3 has been uploaded.
